# Lysosomal membrane glycoproteins bind cholesterol and contribute to lysosomal cholesterol export

Jian Li, Suzanne R Pfeffer*

Department of Biochemistry, Stanford University School of Medicine, Stanford, United States

**Abstract** LAMP1 and LAMP2 proteins are highly abundant, ubiquitous, mammalian proteins that line the lysosome limiting membrane, and protect it from lysosomal hydrolase action. LAMP2 deficiency causes Danon's disease, an X-linked hypertrophic cardiomyopathy. LAMP2 is needed for chaperone-mediated autophagy, and its expression improves tissue function in models of aging. We show here that human LAMP1 and LAMP2 bind cholesterol in a manner that buries the cholesterol 3β-hydroxyl group; they also bind tightly to NPC1 and NPC2 proteins that export cholesterol from lysosomes. Quantitation of cellular LAMP2 and NPC1 protein levels suggest that LAMP proteins represent a significant cholesterol binding site at the lysosome limiting membrane, and may signal cholesterol availability. Functional rescue experiments show that the ability of human LAMP2 to facilitate cholesterol export from lysosomes relies on its ability to bind cholesterol directly.

## Introduction

Eukaryotic lysosomes are acidic, membrane-bound organelles that contain proteases, lipases and nucleases and degrade cellular components to regenerate catabolic precursors for cellular use (*Xu and Ren, 2015*; *Schwake et al., 2013*; *Saftig and Klumperman, 2009*). Lysosomes are crucial for the degradation of substrates from the cytoplasm, as well as membrane bound compartments derived from the secretory, endocytic, autophagic and phagocytic pathways. The limiting membrane of lysosomes is lined with so-called lysosomal membrane glycoproteins (LAMPs) that are comprised of a short cytoplasmic domain, a single transmembrane span, and a highly, N- and O-glycosylated lumenal domain (*Wilke et al., 2012*; *Kundra and Kornfeld, 1999*; *Granger et al., 1990*). Because of their abundance and glycan content, LAMPs have been proposed to serve as a protective barrier to block hydrolase access to the limiting phospholipid bilayer. LAMP1 and LAMP2 are 37% identical and may overlap in function, but knockout of LAMP1 in mouse has a much milder phenotype than depletion of LAMP2 (*Tanaka et al., 2000*): LAMP2-deficient mice have very short lifespans, and show massive accumulation of autophagic structures in most tissues. Indeed, LAMPs are required for fusion of lysosomes with phagosomes (*Huynh et al., 2007*) and LAMP2 has also been proposed to serve as a receptor for chaperone-mediated autophagy (*Cuervo and Dice, 1996*, *2000*; *Bandyopadhyay et al., 2008*).

Previous work has implicated LAMP2 in cholesterol export from lysosomes, as LAMP-deficient cells show cholesterol accumulation that can be rescued by LAMP2 expression (*Eskelinen et al., 2004*; *Schneede et al., 2011*). Proteome-wide analysis of cholesterol binding proteins included LAMP1 and LAMP2 among a long list of candidate proteins (*Hulce et al., 2013*). Despite these hints, the precise function of LAMP proteins has remained unclear, and they are often presumed to be

*For correspondence: pfeffer@ stanford.edu

**eLife digest** Living cells contain many membrane-bound compartments surrounded by a gel-like substance called the cytoplasm. Lysosomes are compartments found in most animal cells, which contain enzymes that can break down virtually all kinds of biological molecules. Cell biologists around the world use two proteins called LAMP1 and LAMP2 to mark lysosomes to study them. The loss of LAMP2 causes a condition called Danon disease that is characterized by thickening of the heart muscle. However, relatively little is known about what these proteins actually do.

Previous studies had hinted that these proteins might bind to the fatty molecule, cholesterol. Li and Pfeffer set out to test this directly and showed that LAMP1 and LAMP2 proteins do indeed bind to cholesterol. The two LAMP proteins also interact with another two proteins, called NPC1 and NPC2, which export cholesterol out of lysosomes.

Li and Pfeffer then showed that cells contain 5- to 10-times more LAMP proteins than they do NPC1-cholesterol exporters. This suggests that LAMP proteins have additional roles that need to be characterized and studied to see how important cholesterol binding is for these processes too. Future studies could also explore if LAMP proteins signal that free cholesterol is available for the cell's needs.

structural components. We show here that LAMP proteins bind cholesterol directly and this capacity contributes to their role in cholesterol export from lysosomes.

## Results and discussion

We sought to verify direct cholesterol binding to LAMP proteins using LAMP protein lumenal domains, engineered to be secreted from cells by simple deletion of their transmembrane and short cytoplasmic domains (*Figure 1—figure supplement 1*; *Figure 1A*). Soluble, purified, LAMP1 and LAMP2 proteins appeared to bind [3]H-cholesterol saturably, at a stoichiometry comparable to equimolar amounts of purified, NPC1 N-terminal domain (NTD) that contains a single cholesterol binding site (*Infante et al., 2008*; *Kwon et al., 2009*; *Figure 1B,D,E*) (note that this does not provide information about relative binding affinities). Binding was not especially sensitive to pH (*Figure 1H*) and was complete after ~2 hr at 4°C (*Figure 1I*).

Cholesterol is poorly soluble, thus binding reactions were carried out in the presence of sub-critical micelle concentration amounts of Nonidet P40 detergent (0.004%) to help solubilize the cholesterol, as worked out by Infante et al. in their studies of cholesterol binding to the N-terminal domain of NPC1 protein (*Infante et al., 2008*). Under these conditions, most of the cholesterol remains in a mixed micelle of cholesterol and detergent and is still poorly soluble. Thus, the apparent affinity for cholesterol is likely to be tighter than the curves indicate, as the amounts added do not reflect the concentration of free cholesterol that is actually available for binding.

Preliminary experiments showed that [3]H-cholesterol binding was competed by unlabeled cholesterol, 24-hydroxycholesterol, but not cholesterol sulfate (*Figure 1G*). This suggested that binding occurs via the 3β-hydroxyl moiety of the cholesterol molecule, similar to the orientation with which the NPC1 N-terminal domain binds cholesterol (*Kwon et al., 2009*). Consistent with this, LAMP2 also bound [3]H-25-hydroxycholesterol with similar apparent affinity as cholesterol; binding was competed by excess cold 25-hydroxycholesterol (*Figure 1C*), as would be expected for a specific interaction. Only low levels of background binding were detected using GFP-binding protein or TIP47 as controls (*Figure 1D,E*). More detailed analysis confirmed that 25-hydroxycholesterol (*Figure 2B*) and 7-ketocholesterol (*Figure 2D*), but not cholesterol sulfate (*Figure 2A*), compete with [3]H-cholesterol for binding to LAMP2 protein.

Epicholesterol is a cholesterol epimer that differs only in the chirality of carbon 3 such that the hydroxyl group is in the alpha rather than beta conformation. Importantly, epicholesterol failed to compete for cholesterol binding to LAMP2 protein under conditions where cholesterol competed for binding (*Figure 2C*). It was not possible to add higher concentrations of sterol competitors due to solubility issues, but significant inhibition was observed. Together, these data strongly support the conclusion that cholesterol binds LAMP2 via its 3β-hydroxyl moiety.

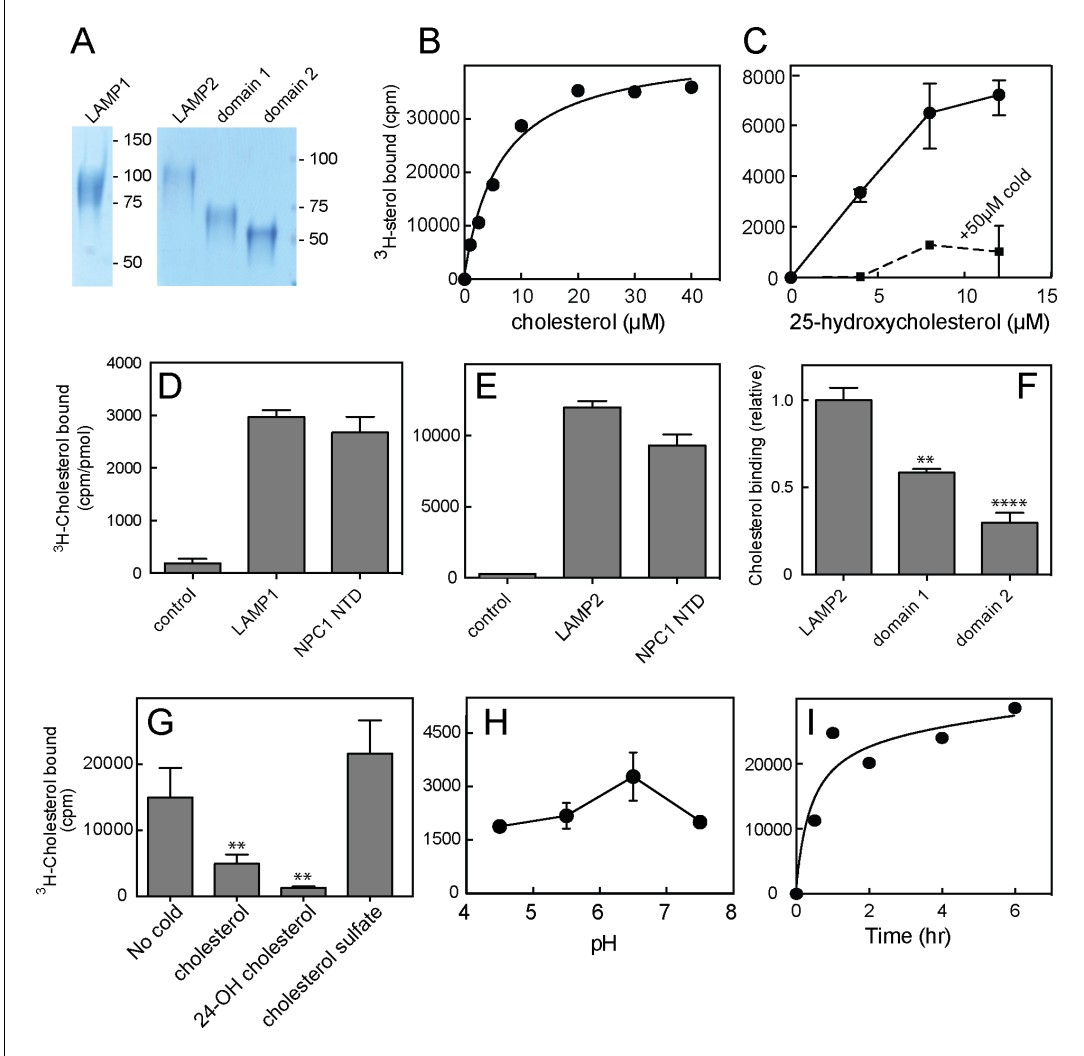

**Figure 1.** Cholesterol binding to LAMP proteins. (**A**) Coomassie-stained SDS-PAGE of purified, secreted human LAMP1, LAMP2 or LAMP domains from LAMP2. **B,C**. $^3$H-cholesterol or $^3$H-25 hydroxycholesterol binding to soluble LAMP2 (full length protein). Also shown in **C** is binding in the presence of 50 µM cold hydroxycholesterol. **D,E**, $^3$H-cholesterol binding to indicated, soluble proteins compared with the soluble NPC1 N-terminal domain. (**F**), Cholesterol binding to LAMP2 domains 1 and 2 compared with full length, soluble LAMP2 protein. P values were determined relative to the full length soluble LAMP2 protein. (**G**) Sterol competition (30 µM) for $^3$H-cholesterol binding to soluble LAMP2. P values were determined relative to no cold addition. **H, I**. pH dependence and kinetics of $^3$H-cholesterol binding to LAMP2. In **B,C,E,H** and **I**, a representative experiment is shown; **C** and **E** show the average of duplicates. In **C**, the background counts in reactions containing the control protein, GFP-binding protein, were subtracted; in **D** and **E**, the control was TIP47 protein. **D,F**, and **G** show the combined results of two experiments in duplicate. Numbers at right (**A**) indicate mass in kD for this and all subsequent figures. Reactions contained **D,E,G,H,I**, 500 nM total cholesterol; **F**, 5 µM total cholesterol; **B** and **C** were carried out using increasing concentrations of the indicated sterol.

The following figure supplement is available for figure 1:

**Figure supplement 1.** Diagram of constructs used.

A slight stimulation of binding was seen in reactions containing low levels of competitor cholesterol sulfate or 25-hydroxycholesterol (*Figure 2A,B*); this is presumably due to the higher solubility of these sterols, which will help solubilize $^3$H-cholesterol present in the reaction's mixed micelles, and presumably make it more available for LAMP2 binding (see also ref. *Infante et al., 2008*). Despite its somewhat higher solubility, 25-hydroxycholesterol did not appear to bind LAMP2 much

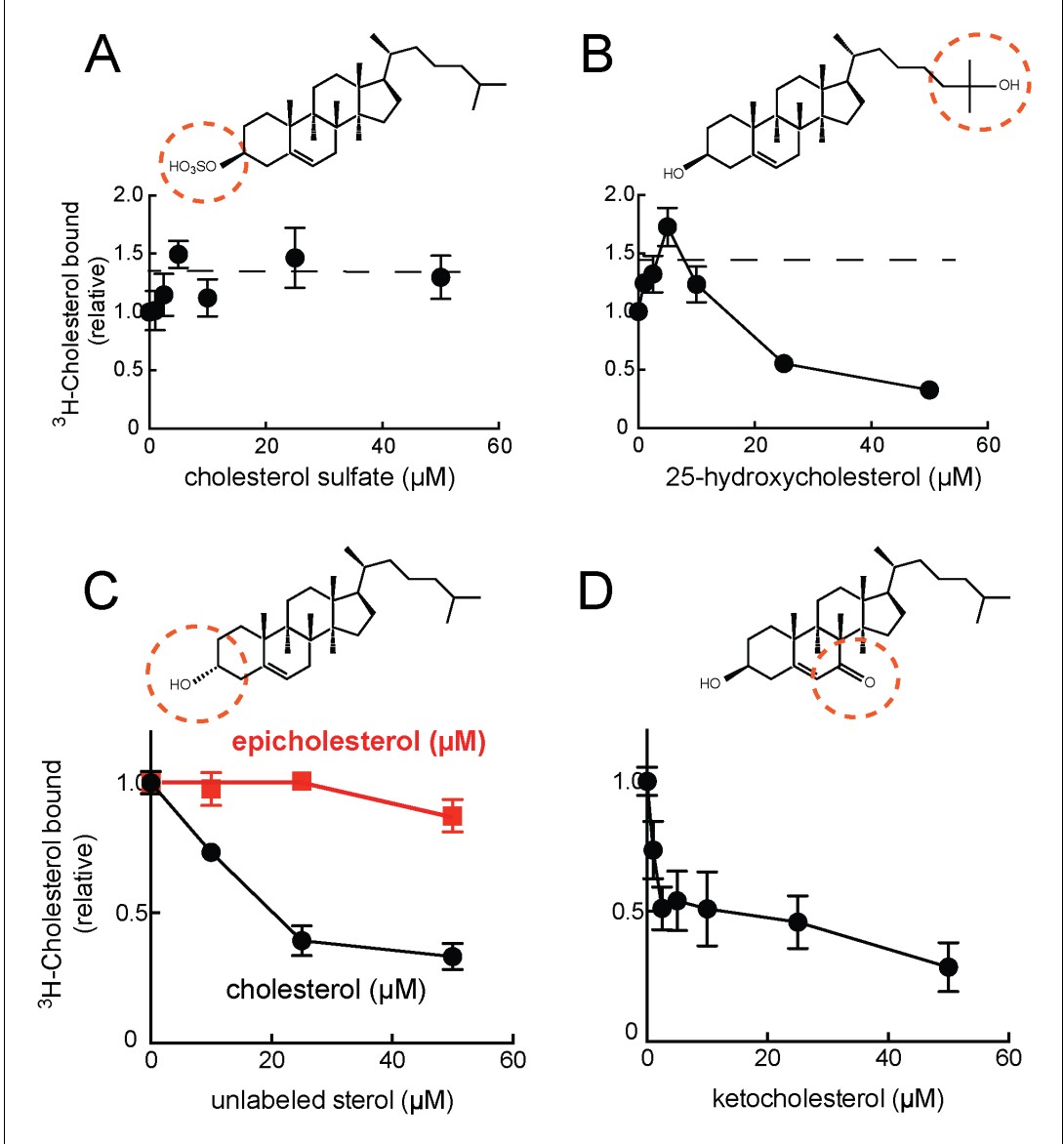

**Figure 2.** Cholesterol binding to LAMP2 is competed by 25-hydroxycholesterol (**B**), cholesterol (**C**), 7-ketocholesterol (**D**) but not cholesterol sulfate (**A**) or epicholesterol (**C**). The structures above indicate the regions of the sterol that differ from cholesterol. In **C**, the background obtained in reactions containing GFP was subtracted. All panels used 50nM $^3$H-cholesterol and the indicated amounts of competitors.

more tightly than cholesterol, at least as inferred from its ability to compete with cholesterol for binding (*Figure 2B*) or to bind directly (*Figure 1C*).

The LAMP protein family includes LAMP1, LAMP2, DC-LAMP, BAD-LAMP and Macrosialin (*Wilke et al., 2012*). Each of these proteins contains a related 'LAMP' domain; LAMP1 and LAMP2 proteins each contain two (*Figure 1—figure supplement 1*). Soluble versions of the individual, membrane distal ('domain 1') and membrane proximal ('domain 2') LAMP domains of LAMP2 (*Figure 1A*) bound cholesterol with different capacity: the N-terminal, domain 1 bound more cholesterol than its membrane proximal, domain 2 counterpart under these conditions (*Figure 1F*). It is possible that both are capable of binding cholesterol within the full length molecule, as total domain 1 binding was less than that seen with the full length, secreted LAMP2 construct (*Figure 1F*).

To verify that LAMP2 binds cholesterol in cells, soluble LAMP2 protein was expressed and purified from the secretions of HEK293F cells grown in protein-free, FreeStyle 293 Expression Medium

that does not contain cholesterol. Under these conditions, any LAMP2-bound sterol must come from intracellular sources. We subjected freshly purified LAMP2 protein to chloroform:methanol extraction and analyzed the extract by thin layer chromatography. As shown in *Figure 3C*, the LAMP2 extract contained a molecular species that co-chromatographed with cholesterol but not 24-hydroxy-, 25-hydroxy-, or 26-hydroxycholesterol, lanosterol or 7-beta-hydroxycholesterol. Mass spectrometry of the eluted material (*Figure 3B*) confirmed a profile identical with purified cholesterol standard (*Figure 3A*) (*Figure 3C*). These experiments show that LAMP2 purified from cell secretions carries primarily, bound cholesterol.

NPC1 and NPC2 proteins mediate cholesterol export from lysosomes (*Kwon et al., 2009*; *Rosenbaum and Maxfield, 2011*). NPC1 has 13 transmembrane domains, and three large, lumenal domains that are important for its function. As mentioned earlier, the NPC1 N-terminal domain binds cholesterol directly (*Kwon et al., 2009*). Because of a possible connection between LAMP protein cholesterol binding and NPC-mediated cholesterol export, we checked for an interaction between these proteins. Membrane anchored, endogenous LAMP2 co-immuno-precipitated with full length NPC1-GFP but not with the control protein, GFP (*Figure 4A*) or the lysosomal membrane protein, MCOLN1 (*Figure 4D*), upon expression in HEK293T cells. Interestingly, co-

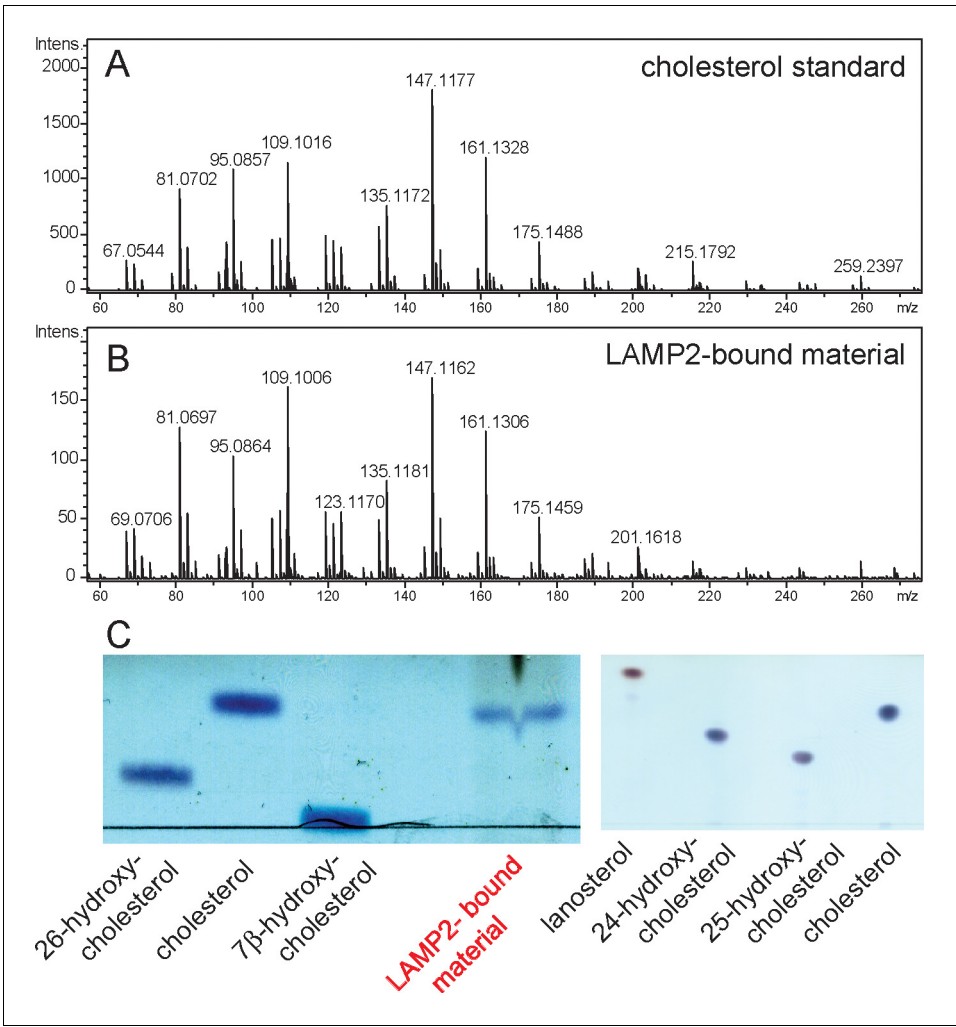

**Figure 3.** Mass spectrometry identification of small molecules released from LAMP2 after chloroform:methanol (2:1) extraction. (**A**) masses from cholesterol standard; (**B**) masses of LAMP2-bound material; (**C**) Copper sulfate/phosphoric acid detection of indicated markers after thin layer chromatography compared with material eluted from soluble LAMP2 (50 μg). Shown are the results of a representative experiment carried out twice.

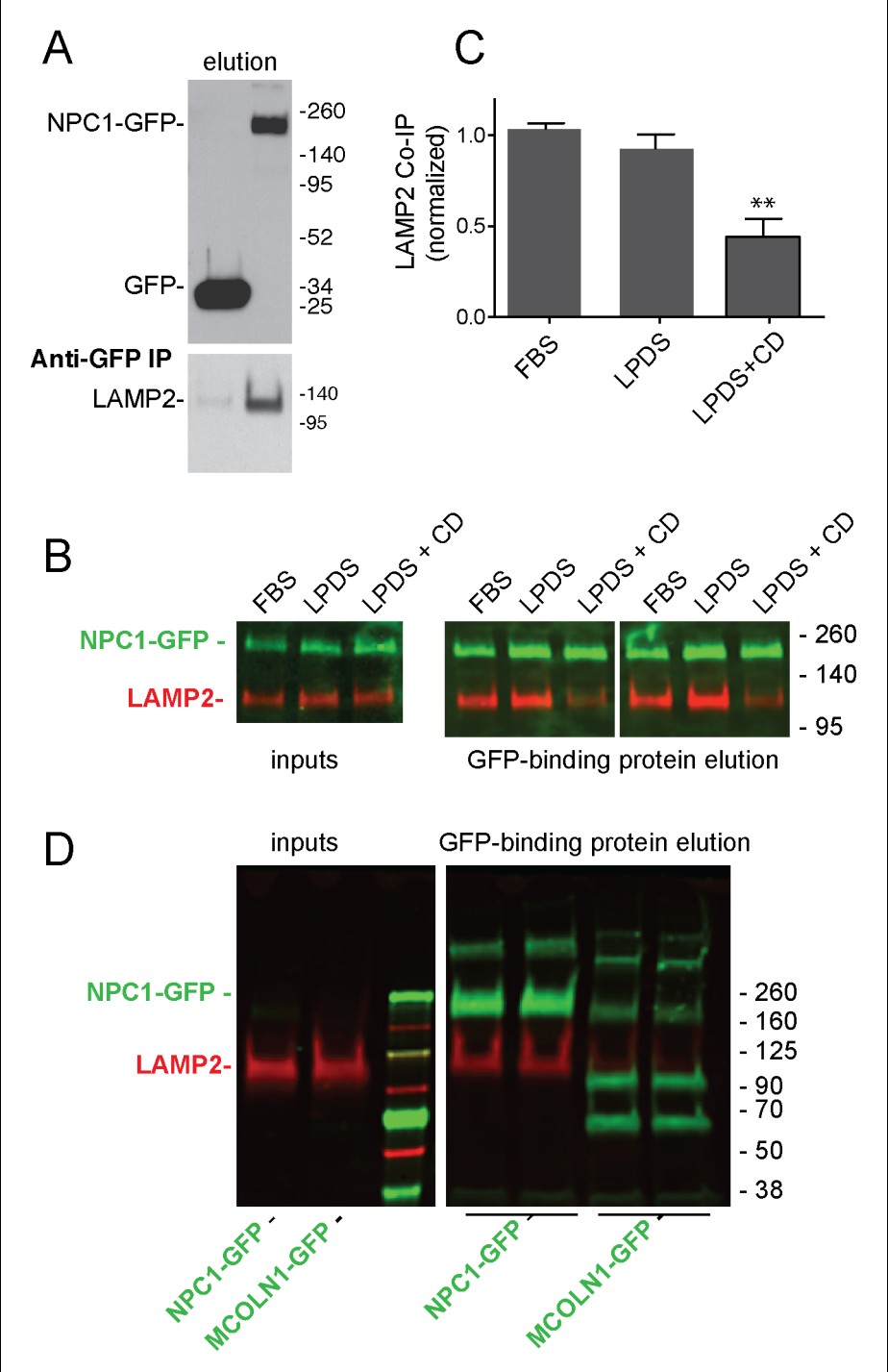

**Figure 4.** LAMP2 interacts with NPC1 protein. (**A**) Anti-GFP immunoprecipitation from HEK293T cells grown in FBS containing medium expressing GFP or mouse NPC1-GFP. The blot was developed with ECL. Upper panel, anti-GFP immunoblot (100% elution); lower panel, anti-LAMP2 immunoblot to detect endogenous, full length protein (100% elution). **B,C**, co-immunoprecipitation of LAMP2 and NPC1-GFP in cells grown in FBS, LPDS (5%) or LPDS + cyclodextrin (1 mM) for 24 hr. Left panels, 1% inputs; right panels, 50% elutions. Error bars represent SEM for two combined experiments carried out in duplicate; P value is from comparison with LPDS by two-tailed Student's t-test. **B** shows duplicate reactions to document reproducibility. **D**, Anti-GFP immunoprecipitation of HEK293T cells grown in FBS expressing GFP-MCOLN1 or mouse NPC1-GFP. GFP-MCOLN1 occurs as a ~90 kD form, a proteolytically processed form, and as higher oligomers (***Vergarajauregui et al., 2011***). Left panels, inputs (2%);

*Figure 4 continued on next page*

*Figure 4 continued*

right panels, total elution carried out in duplicate. GFP proteins are green; LAMP2 is presented in red. Shown is a representative experiment carried out twice in duplicate.

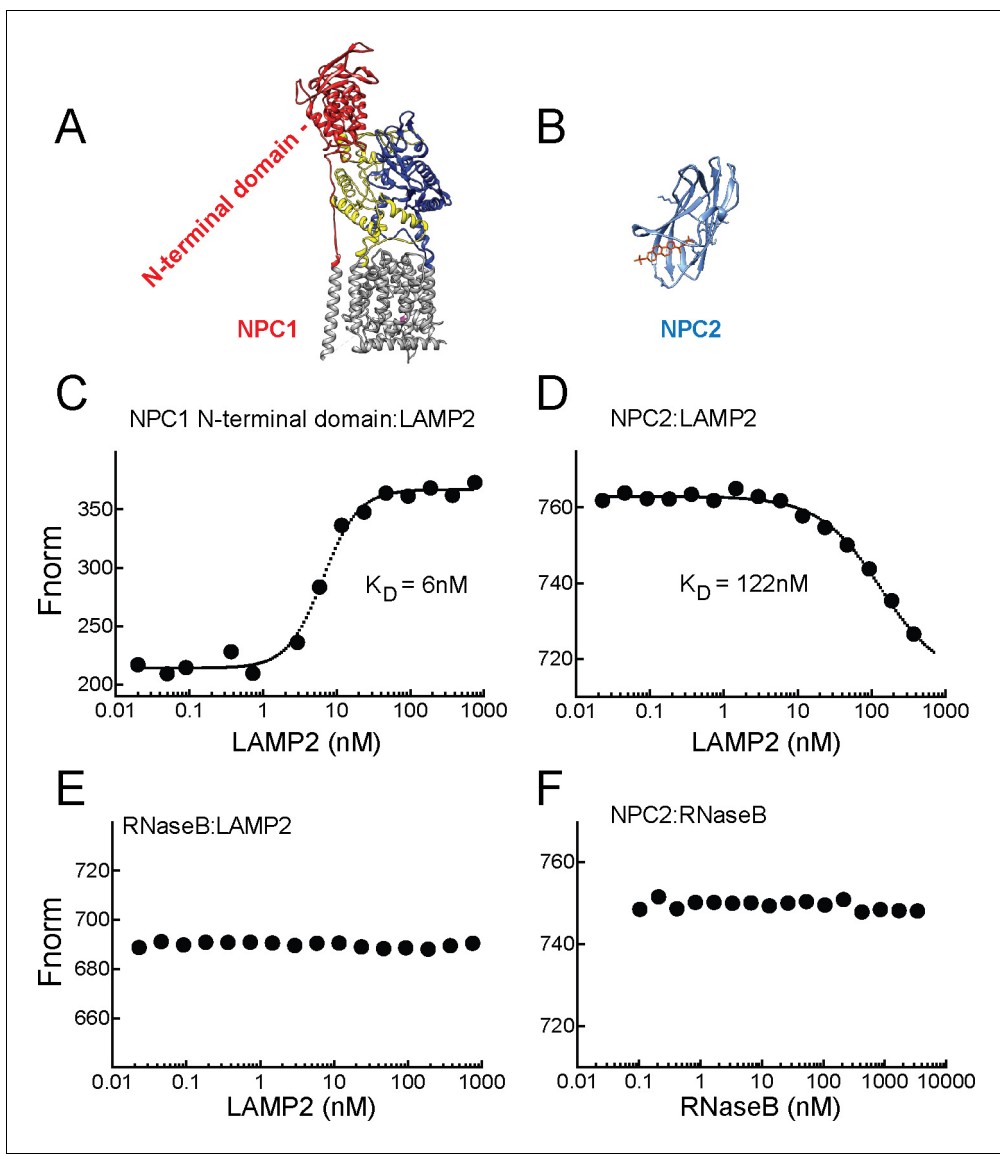

**Figure 5.** LAMP2 binds NPC1 and NPC2 proteins. **A,B**, Structures of NPC1 (*Gong et al., 2016*; pdb 3jD8) and NPC2 (*Xu et al., 2007*; pdb 2 hka) proteins; **C,E**, Microscale thermophoresis (**E**) or fluorescence (**C**) obtained with mixtures of soluble, AF647 labeled-NPC1 N-terminal domain or AF647-RNase B with increasing concentrations of soluble LAMP2 in 1 μM cholesterol. **D,F**, Microscale thermophoresis of AF647 labeled-NPC2 with increasing concentrations of soluble LAMP2 or RNase B in 1 μM cholesterol sulfate. For **C–F**, a representative experiment carried out at least twice is shown.

The following figure supplement is available for figure 5:

**Figure supplement 1.** Quantitation of NPC1 (**A**) and LAMP2 (**B**) proteins in HeLa and HEK293T cells.

immunoprecipitation decreased in cells treated for 24 hr with cyclodextrin to remove cholesterol from lysosomes (*Abi-Mosleh et al., 2009*; *Rosenbaum et al., 2010*) and the plasma membrane (*Figure 4B,C*). These conditions (~0.1% cyclodextrin) have been shown to be non-toxic (cf. *Ulloth et al., 2007*) and did not alter cell growth rate or viability in our hands.

Purified, soluble LAMP2 protein also bound very tightly (and directly) to the N-terminal domain of NPC1 protein (*Figure 5A*, red) in the presence (or absence, not shown) of cholesterol ($K_D = 6$ nM), as monitored by microscale thermophoresis using AF647 dye-conjugated NPC1 protein—binding significantly altered the fluorescence of NPC1 protein (*Figure 5C*). No interaction was observed for the control glycoprotein, RNase B (*Figure 5E*). The smaller, NPC2 protein (*Figure 5B*) also bound to LAMP2 ($K_D = 122$ nM), but ~20 fold less tightly that NPC1 N-terminal domain (*Figure 5D*); binding was monitored in the presence of cholesterol sulfate which will occupy the binding site of NPC2 but not LAMP2 (*Figures 1* and *2*). No binding was seen for NPC2 to the control RNase B protein (*Figure 5F*). Thus, LAMP2 binds directly to both NPC1 N-terminal domain and NPC2 proteins. For NPC1, the enhanced binding seen in cells in the presence of cholesterol does not appear to reflect occupancy of NPC1's N-terminal domain, as this variable did not influence LAMP2 binding in solution. However, it is important to note that NPC1 also likely binds cholesterol within its membrane spanning region which may also influence its overall conformation (*Lu et al., 2015*; *Li et al., 2016*).

LAMP proteins are highly abundant components of the lysosome membrane and may serve as a reservoir for cholesterol extracted from intra-lysosomal membranes by NPC2, prior to cholesterol export from lysosomes by NPC1. [The term, reservoir, is meant to imply a holding station for cholesterol molecules that have been solubilized from the internal lipid contents of lysosomes by NPC2, and held closer to the limiting membrane, prior to NPC1-mediated export.]

Are LAMP proteins abundant enough to represent a cholesterol reservoir? We used purified LAMP2 and NPC1 proteins as standards to determine their precise abundance in HeLa and HEK293 cell lysates (*Figure 5—figure supplement 1*). Using the polypeptide molecular weights and cellular protein determinations, we estimate that HeLa cells contain $6.8 \times 10^6$ LAMP2 molecules per cell and $3.7 \times 10^5$ NPC1 molecules per cell, or 18 fold more LAMP2 than NPC1; HEK293T cells contain $2 \times 10^6$ LAMP2 molcules per cell and $5.9 \times 10^5$ NPC1 molecules (3.6 × fold more LAMP2).

Baby hamster kidney cells have been estimated to contain an absolute volume of ~37 $\mu m^3$ lysosomes and prelysosomes per cell ($3.7 \times 10^{-14}$ l) and a lysosome membrane area of 370 $\mu m^2$ (*Griffiths et al., 1989*). Assuming similar values for HeLa cells, this would represent a LAMP2 membrane density of 18,378 or 5676 molecules per $\mu m^2$ in HeLa or 293T cell lysosomes, respectively, consistent with previous reports (*Granger et al., 1990*). For comparison, tightly packed viral spike glycoproteins occur at a density of 22,000 molecules per $\mu m^2$ (*Quinn et al., 1984*). We assume that LAMP1 will be of similar high density, together with LAMP2, practically lining the interior of the lysosome limiting membrane. In terms of concentrations, 37 femtoliters of lysosome volume would contain 0.3 mM LAMP2-associated cholesterol binding sites in HeLa cell lysosomes (assuming one mole cholesterol bound per more LAMP2). It does not seem unreasonable to consider this as a significant reservoir of cholesterol molecules that may be poised for transfer to NPC1 protein prior to export.

## Residues needed for cholesterol binding are needed for LAMP protein function

The structure of an individual LAMP domain from DC-LAMP protein is comprised of a novel, beta-prism fold that appears to contain a hydrophobic pocket (*Wilke et al., 2012*); we used this structure to model the structure of LAMP2 domain 1 (*Figure 6A*). Site directed mutagenesis of hydrophobic residues predicted to line the walls of this cavity yielded purified LAMP2 proteins with impaired cholesterol binding activity. Thus, a soluble, LAMP2 domain 1-$I^{111}A/V^{114}A$ construct yielded a secreted protein (*Figure 6B* inset, right lane) that bound significantly less cholesterol than its wild type counterpart (*Figure 6B* inset, left lane and panel B). Because these proteins were obtained from cell secretions, they are likely to be properly folded, as they escaped the endoplasmic reticulum's quality control machinery. These experiments show that residues facing the predicted, prism fold pocket are important for cholesterol binding and likely contribute to the cholesterol binding site.

Finally, to verify the importance of cholesterol binding to LAMP2 protein as part of its physiological role, we tested the ability of wild type and mutant LAMP2 constructs to rescue the cholesterol accumulation seen in lysosomes from mouse embryonic fibroblasts missing LAMP1 and LAMP2 proteins (*Eskelinen et al., 2004*; *Schneede et al., 2011*). The ability of lysosomes to export cholesterol

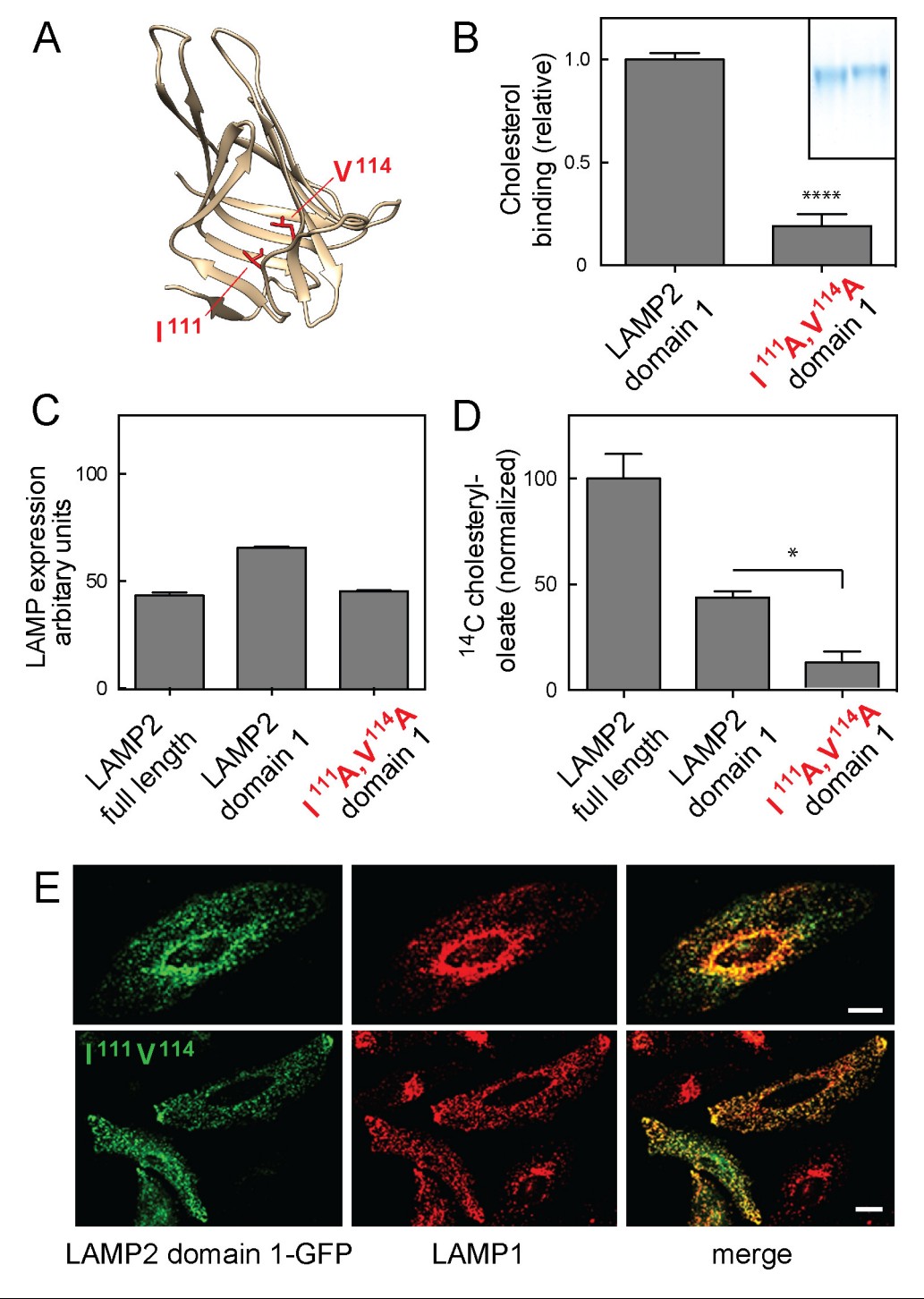

**Figure 6.** Cholesterol binding to LAMP2 domain 1 is required for its ability to rescue cholesterol export from LAMP-deficient lysosomes. (**A**), predicted structure model of LAMP2 domain 1; residues I111 and V114 are highlighted in red. (**B**) Relative $^3$H-cholesterol binding to soluble LAMP 2 domain 1 or LAMP2 domain 1-I$^{111}$A/V$^{114}$A. Shown is combined data from 5 independent experiments carried out in duplicate in the presence of 50 nM $^3$H-cholesterol. Inset, SDS-PAGE analysis of wild type (left) and domain 1-I$^{111}$A/V$^{114}$A (right) proteins analyzed. P value was determined by two-tailed Student's t-test. (**C**) flow cytometry analysis of mean fluorescence of GFP rescue constructs in lentivirus-tranduced cells (>20,000 cells analyzed). (**D**) Cholesteryl oleate synthesis in MEF cells lacking LAMP1 and LAMP2 after rescue with either full length, membrane anchored LAMP2, membrane anchored LAMP2 domain 1, or membrane anchored LAMP2 domain 1-I$^{111}$A/V$^{114}$A. C-terminally GFP-tagged, rescue proteins

*Figure 6 continued on next page*

*Figure 6 continued*

were stably expressed using lentivirus transduction; shown is the combined result of 2 independent experiments, normalized for the amount of mature protein in each sample (*Figure 6—figure supplement 1*) relative to the amount of rescue seen with full length LAMP2 protein. P-values are in relation to full length for domain 1, or to domain 1 for the mutant protein, and were determined by one way ANOVA. E, confocal light microscopic analysis of GFP rescue construct localization (green) and endogenous LAMP1 protein (red) in transiently transfected HeLa cells; bars represent 20 µm.

The following figure supplement is available for figure 6:

**Figure supplement 1.** Immunoblot analysis of LAMP2 constructs from lentivirus transduced LAMP1/LAMP2-knockout MEF cells.

can be monitored by feeding cells cholesterol in the form of LDL, and using conversion of $^{14}$C-oleic acid to cholesteryl oleate that takes place after endocytosed cholesterol is transported to the endoplasmic reticulum (*Goldstein et al., 1983*). Previous work showed that LAMP1/2 knockout MEF cells were impaired in cholesterol export using this assay (*Schneede et al., 2011*).

We used lentivirus transduction to test the ability of full length LAMP2, a membrane anchored LAMP2 domain 1 (LAMP2-GFP Δ194–368; *Figure 1—figure supplement 1*), or a membrane anchored LAMP2 domain 1-I$^{111}$A/V$^{114}$A to rescue the ability of LAMP1/LAMP2 knockout MEF cells to export LDL-derived cholesterol from lysosomes. For these experiments, we used LAMP constructs containing a single LAMP domain, as full length LAMP2 constructs with mutations in both LAMP domains failed to fold properly or be transported efficiently to lysosomes.

It was important to first verify the precise amounts of each construct in lysosomes, to evaluate any functional rescue findings. Flow cytometry analysis showed that the rescue constructs were expressed at comparable levels in each stably expressing cell population (*Figure 6C*). Light microscopy confirmed that the constructs were capable of proper lysosome localization, as determined by their colocalization with endogenous LAMP1 protein (*Figure 6E*) in HeLa cells. (Similar staining was observed in LAMP knockout MEF cells that lack LAMP protein markers).

To fully confirm the folding of these artificial constructs, we analyzed their glycosylation status and stability after addition of cycloheximide to inhibit new protein synthesis (*Figure 6—figure supplement 1*). Full length GFP-LAMP2 protein migrated at ~140 kD and its abundance was not altered after 4 hr cycloheximide treatment, consistent with its long half life in cultured cells (panel B). Similarly, the GFP-domain 1 construct was stable under these conditions and migrated at ~90 kD (panels A,B). In contrast, the I$^{111}$A/V$^{114}$A mutant domain I protein displayed two distinct bands; the upper band was stable, while the lower band likely corresponded to an ER form that was largely degraded after 4 hr in cycloheximide (panels A,B). From this we conclude that cells expressing membrane anchored LAMP2 domain 1 I$^{111}$A/V$^{114}$A are less efficient at folding the protein but some folded protein makes it to lysosomes, where it is stable. This difference was accounted for in subsequent functional rescue experiments (*Figure 6D*).

*Figure 6D* shows that as expected, full length, wild type LAMP2 rescued cholesterol export in LAMP1/2-deficient MEF cells; membrane anchored LAMP2 domain 1 showed a level of rescue consistent with its lower capacity for cholesterol binding (cf. *Figure 1F*). Importantly, membrane anchored, LAMP2 domain 1 I$^{111}$A/V$^{114}$A failed to rescue cholesterol export from lysosomes (*Figure 6D*), consistent with its inability to bind cholesterol; shown are the data corrected for the amount of mature proteins present in lysosomes in these cells. LAMP2 constructs mutated in both cholesterol-binding sites could not be tested, as they were only poorly delivered to lysosomes.

These experiments demonstrate a direct role for LAMP2 in cholesterol export from lysosomes, and confirm that LAMP2's ability to bind cholesterol correlates with its ability to support cholesterol export from LAMP-deficient MEF cells. In addition, LAMP proteins bind tightly to NPC proteins in vitro and in cells, and appear to facilitate cholesterol export from lysosomes.

LAMP proteins are the most highly abundant membrane glycoproteins of the lysosome, and their lumenally oriented cholesterol binding sites represent a significant binding site for this important sterol. We measured ~7 × 10$^6$ molcules per HeLa cell, representing 0.3 mM binding sites in lysosomes. A recent cellular mass spectrometry analysis (*Itzhak et al., 2016*) estimated LAMP proteins

to be present at 260,000 copies and NPC1 at 29,193 copies per HeLa cell. While the relative abundance of these proteins matches the values we report here, their total level was 25 fold lower in that study. It is possible that these transmembrane glycoproteins were under-represented in due to their unusual protease resistance as proteins of the lysosome membrane, differences in cell confluency and/or differences in HeLa cell lines employed.

Lysosomes have recently been shown to sense and signal amino acid availability to influence lysosome biogenesis in relation to cellular need (*Settembre et al., 2013*), and LAMP oligomerization has been reported to correlate with chaperone mediated autophagy (*Bandyopadhyay et al., 2008*). Cholesterol levels may influence LAMP protein conformation or interaction with other partners to signal the availability of endocytosed cholesterol to influence autophagy and cellular metabolism. The ten fold higher abundance of LAMP proteins compared with NPC1 protein in HeLa cells suggests that LAMP proteins may do more than just facilitate NPC1 function in cholesterol export. Future experiments will be needed to fully understand the roles played by these highly abundant lysosomal membrane glycoproteins.

We have shown that LAMP2 binds tightly to the N-terminal domain of NPC1 and also binds cholesterol with the same orientation as that domain. LAMP2 also aids in cholesterol export from lysosomes. How might LAMP2's cholesterol binding site contribute to cholesterol export? Current models suggest that the soluble NPC2 protein binds cholesterol from the internal membranes of lysosomes and delivers it to NPC1 at the limiting membrane of this compartment (*Kwon et al., 2009*). One possibility is that NPC2 can deliver cholesterol to both NPC1 and to LAMP2, which is more abundant. This would help drive the cholesterol export process by moving cholesterol from the accumulated, lumenal lipid stores to the lysosome's limiting membrane. Because LAMP2 and NPC1 N-terminal domains bind cholesterol in the same orientation, it makes sense that NPC2 (which binds in opposite orientation, [*Xu et al., 2007*]) could transfer the cholesterol between these two proteins. The recent crystal structure of NPC2 bound to the middle, lumenal domain of NPC1 (*Li et al., 2016*) supports a direct handoff between NPC1 and NPC2 (*Kwon et al., 2009*). In future work, it will be important to elucidate precisely how LAMP2 interacts with both NPC2 and NPC1 to facilitate cholesterol export from lysosomes and how cholesterol binding contributes to LAMP2's other cellular roles.

## Materials and methods

Cholesterol, epicholesterol and sodium cholesteryl sulfate were from Sigma (St. Louis, MO); 24-hydroxycholesterol (24-HC) was a gift from Rajat Rohatgi (Stanford University, Stanford, CA); 25-hydroxycholesterol and 7-ketocholesterol were from Steraloids (Newport, RI) or Avanti Polar Lipids (Alabaster, AL); [1,2-$^3$H]cholesterol (50 Ci/mmol) and 25-[26,27-$^3$H] hydroxycholesterol were from American Radiolabeled Chemicals (St. Louis, MO). Ni-NTA agarose was from Qiagen (Valencia, CA); freestyle 293 expression medium and Dulbecco's modified Eagle's medium (DMEM) was from Life Technologies (Carlsbad, CA); lipoprotein deficient serum was from KALEN Biomedical (Montgomery Village, Maryland). Pierce Protein Concentrators PES were from Thermo Fisher Scientific (Grand Island, NY); PD-10 desalting columns and Q-Sepharose were from GE Healthcare Life Sciences (Pittsburgh, PA); pFastBac NPC1-N-terminal domain plasmid, LAMP1-mGFP and MCOLN1-pEGFP C3 were from Addgene (Cambridge, MA); pGEM-LAMP2 was from Sino Biological Inc; mouse anti-human LAMP1 and LAMP2 antibody culture supernatants were from Developmental Studies Hybridoma Bank (University of Iowa, Iowa city, IA). Rabbit monoclonal anti-NPC1 was from AbCam (Cambridge, MA); Chicken anti-GFP antibody was from Aves Labs (Tigard, Oregon); IRDye 800CW donkey anti-chicken and IRDye 680RD donkey anti-mouse antibodies were from LI-COR, Inc. (Lincoln, NE); anti-chicken-HRP conjugate was from Promega (Sunnyvale, CA); goat anti-mouse-HRP conjugate was from BioRad (Hercules, CA); ECL Western Blotting Substrate was from Thermo Scientific (Rockford, IL).

### Buffers

Buffer A: 50 mM ammonium acetate, pH4.5, 150 mM NaCl, 0.004% NP-40; buffer B: 50 mM MES, pH5.5, 150 mM NaCl, 0.004% NP40; buffer C: 50 mM MES, pH6.5, 150 mM NaCl, 0.004% NP-40; buffer D: 50 mM HEPES, pH7.5, 150 mM NaCl, 0.004% NP-40; buffer E: 25 mM Tris, pH7.4, 150 mM NaCl; RIPA buffer: 50 mM Tris, pH7.4, 150 mM NaCl, 1% NP-40, 0.2% deoxycholic acid, 0.1% SDS.

## Plasmids

cDNAs encoding full length, soluble human LAMP1(1–382), human LAMP2 (1–375) and domain 1 of human LAMP2 (1–231) were PCR amplified from LAMP1-mGFP and pGEM-LAMP2 respectively. The PCR products were inserted into pEGFP-N3 vector. The constructs were assembled to have an unstructured GSTGSTGSTGA linker at the C terminus, followed by a $His_{10}$ tag and a FLAG tag. For LAMP2, another $His_{10}$ tag was added downstream of the FLAG tag for improved purification. LAMP2 domain 2 was prepared by deleting residues 39–219 from the full length, soluble domain construct. FUGENE6 was used for transient transfection of HeLa cells. Membrane anchored rescue constructs were stably expressed in LAMP1/2 deficient MEF cells by lentivirus transduction and were comprised of full length LAMP2 bearing a C-terminal GFP (LAMP2-GFP), or LAMP2-GFP Δ194–368 (encoding membrane anchored domain 1) or the latter construct carrying point mutations.

## Cell lines

Authenticated HEK293F, HEK293T, and HeLa cells were from ATCC and used at low passage; Sf9 cells were purchased from Thermo Fisher Scientific (Waltham, MA); Mouse embryonic fibroblasts from LAMP1/LAMP2 double knockout mice (*Bandyopadhyay et al., 2008*; *Eskelinen et al., 2004*) were the generous gift of Dr. Paul Saftig (Christian-Albrechts-Universität Kiel, Germany). Mycoplasma contamination was monitored by DAPI staining.

## Cell culture

All cells were cultured at 37°C and under 5% $CO_2$ in Dulbecco's modified Eagle's medium supplemented with 7.5% fetal bovine serum, 100 U/ml penicillin and 100 µg/ml streptomycin, unless indicated. HEK293F suspension cells were cultured at 37°C under 5% CO2 in Freestyle 293 medium. In some experiments, cells were cultured in lipoprotein deficient serum (5%).

## Protein purification

pFastBac NPC1-N-terminal domain plasmid was used to make virus for infection of Sf9 insect cells. 72 hr after infection, Sf9 cultures were spun down and ammonium sulfate added to achieve 60% saturation. The resulting precipitate was re-suspended buffer E and incubated with Ni-NTA resin overnight at 4°C. After washing with buffer E with 25 mM imidazole, the protein was eluted with buffer E plus 250 mM imidazole, and further purified using Q-Sepharose.

HEK293F cells were transfected using 293fection according to the manufacturer. After 72 hr, supernatants were collected after spinning 3000 rpm for 5 min. To purify proteins for [3H] cholesterol binding, supernatants were subjected to 90% ammonium sulfate precipitation. After spinning at 13,000 rpm for 30 min, pellets were re-suspended in buffer E plus 25 mM imidazole and incubated with Ni-NTA resin overnight at 4°C, followed by washing with the same buffer. Bound proteins were eluted with buffer E plus 250 mM imidazole. Proteins were concentrated and buffer exchanged into buffer C with Pierce Protein Concentrators PES (10 kD cut-off). Proteins were either used immediately or stored at −80°C after snap freezing in liquid nitrogen. For cholesterol extraction and thin layer chromatography, supernatants were adjusted to pH 7.4 and incubated with Ni-NTA resin overnight at 4°C; after washing with buffer E plus 25 mM imidazole, proteins were eluted with buffer E plus 250 mM imidazole. Proteins were desalted into PBS using a PD-10 column.

## 3H-cholesterol binding

Each reaction was carried out in a final volume of 80–100 µl of buffer A, B, C or D containing 0.1–1 µg purified His-tagged protein, 1µg BSA and 10–400 nM 3H-cholesterol diluted with 0.1–50 µM cholesterol. For competition assays, reactions were in 80 µl buffer C (50 mM MES, 150 mM NaCl, 0.004% NP-40, pH6.5) with 0.1 µg full length soluble LAMP2 protein and 50 nM 3H-cholesterol, competition was started by adding vehicle (ethanol) or different concentrations of competitors as indicated. After incubation overnight at 4°C, the mixture was loaded onto a column packed with 30 µl Ni-NTA agarose beads. After incubation for 10 min, each column was washed with 5 ml of buffer C plus 10 mM imidazole. The protein-bound 3H-cholesterol was eluted with 250 mM imidazole-containing buffer C and quantified by scintillation counting.

## Mass spectrometry

Samples were analyzed by LC/MS on an Agilent 1260 HPLC and Bruker microTOF-Q II mass spectrometer. Full scan mass and product ion spectra were acquired in positive ion mode, using a Phenomenex Kinetex C18 2.6u 2.1 × 100 mm column, and an initial condition of 30%, 0.1% formic acid in water/70% methanol.

## Thin layer chromatography

Full length soluble LAMP2, and domains 1 and 2 of LAMP2 were purified as described above. Extraction was performed by adding 3 sample volumes of chloroform/methanol (2:1, v/v) to the samples. After repeating once more, extracts were pooled and dried under nitrogen. The extracts were re-dissolved in 50–100µl chloroform/methanol (2:1, v/v). Samples were spotted onto a Silica gel plate. The plate was developed with isopropanol until the front reached 1cm above the loading position; after drying under airflow, the plate was further developed using 2% methanol in chloroform until the front reached the top of the plate. The plate was sprayed with 10% $CuSO_4$ in 4% or 8% phosphoric acid and heated at 180°C to visualize the samples.

## Co-immunoprecipitation

HEK293T cells expressing pEGFP-N1, pEGFP-N1-mNPC1 or pEGFP-C3-MCOLN1 were harvested 24–48 hr post-transfection and lysed in lysis buffer (50 mM MES, pH 5.5, 150 mM NaCl and 0.1% digitonin) supplemented with protease inhibitors. After 30 min on ice, lysates were spun at 15,000 g for 15 min, and protein concentrations of the supernatants were measured. Equal amounts of extract protein were incubated with GFP-binding protein–conjugated agarose for 2 hr at 4°C. Immobilized proteins were washed 4 times with 1ml lysis buffer, eluted with 2× SDS loading buffer, and subjected to BioRad Mini-PROTEIN TGX 4–20% gradient gels. After transfer to nitrocellulose membrane and antibody incubation, blots were detected with ECL western blotting detection substrate or visualized using LI-COR Odyssey Imaging System.

## Microscale thermophoresis (MST)

MST experiments were performed on a Monolith NT.115Pico instrument (Nanotemper Technologies). Briefly, $His_6$-NPC1 N-terminal domain, RNase B or NPC2 were labeled using the RED-NHS (Amine Reactive) Protein Labeling Kit (Nanotemper Technologies). A constant concentration of 6 nM labeled protein was mixed with binding partnerswith a final buffer condition of 50 mM MES, pH 5.5, containing 150 mM NaCl, 0.004% NP-40. Premium coated capillaries contained 16 sequential, 2 fold serial dilutions. Analysis was at 40% laser power for 30 s, followed by 5 s cooling. Data were normalized to fraction of bound (0 = unbound, 1 = bound). The dissociation constant $K_D$ was obtained by plotting the normalized fluorescence $F_{norm}$ against the logarithm of the different concentrations of the dilution series according to the law of mass action.

## Quantitation of intracellular LAMP2 and NPC1 protein

HeLa and HEK293T cells were grown to sub-confluence in DMEM supplemented with 7.5% FBS. One 10cm dish of cells was washed 3 times with cold PBS, then lysed with 500µl RIPA buffer with protease inhibitor cocktail (Sigma). After 30 min on ice, the lysate was centrifuged at 13,000 rpm for 15 min at 4°C. The resulting supernatant was transferred to a new tube and protein was measured by BCA assay. Lysates were resolved by SDS-PAGE, using different amounts of purified human LAMP2 or NPC1 protein as standards. After transfer to nitrocellulose, the blot was probed with anti-human LAMP2 or NPC1 antibody followed by IRDye 800CW labeled anti mouse (for LAMP2) or rabbit (for NPC1) secondary antibody, and visualized using a LI-COR Odyssey Imaging System and analyzed using ImageJ software. Calculations were based on molecular weights of 45,874 for LAMP2 and 142167 for NPC1 polypeptide chains, and neglected glycan contribution, which is not measured in the protein assay employed. Purified, full length NPC1 protein was the gift of Dr. Xiaochun Li (Rockefeller University) and was N-glycanase treated.

## Other methods

Confocal immunofluorescence microscopy was carried out as described (*Li et al., 2015*). Cells grown on coverslips were fixed with 3.7% (vol/vol) paraformaldehyde for 15 min at room temperature.

LAMP1 staining was performed with sequential incubation of mouse anti-LAMP1 culture supernate and Alexa Fluor 594 goat anti-mouse antibody (1:1,000, Invitrogen), each for 1 hr at room temperature. Coverslips were mounted using Mowiol and imaged using a Leica SP2 confocal microscope and Leica software with a 60 × 1.4 N.A. Plan Apochromat oil immersion lens and a charge-coupled device camera (CoolSNAP HQ, Photometrics). Flow cytometry was carried out on a FACScan Analyzer on gently trypsinized cells fixed as described above (*Li et al., 2015*). Structures were presented in drawings created using Chimera software (*Pettersen et al., 2004*).

### Cholesterol ester formation

LAMP1/LAMP2-deficient MEF cells were cultured in DMEM medium with 5% (vol/vol) LPDS for two days and assayed (*Goldstein et al., 1983*; *Li et al., 2015*) with minor modification. After 48 hr, 100 µg/mL LDL, 50 µM lovastatin, and 50 µM sodium mevalonate were added for 5 hr. Cells were pulse labeled for 4 hr with 0.1 mM sodium [1-$^{14}$C]oleate (American Radiolabeled Chemicals)–albumin complex. Cells were washed two times with 2 mL 50 mM Tris, 150 mM NaCl, 2 mg/mL BSA, pH 7.4, followed by 2 mL 50 mM Tris, 150 mM NaCl, pH 7.4. Cells were extracted and rinsed with hexane-isopropanol (3:2), pooled, and evaporated. After resuspending each sample in 60 µl hexane, 4 µL of lipid standard containing 8 µg/mL triolein, 8 µg/mL oleic acid, and 8 µg/mL cholesteryl oleate was added. Samples were spotted onto a silica gel 60 plastic backed, thin layer chromatogram and developed in hexane. Cholesteryl oleate was identified with iodine vapor, scraped from chromatograms, and radioactivity determined by scintillation counting in 10 mL Biosafe II.

### Statistical analysis

Minimum sample sizes were determined assuming 5% standard error and >95% confidence level. p values were determined using Graphpad Prism software and are indicated in all figures according to convention: $*p \leq 0.05$; $**p \leq 0.01$; $***p \leq 0.001$; $****p \leq 0.0001$. Error bars represent standard error of the mean.

## Acknowledgements

This research was funded by a grant from the Ara Parseghian Medical Research Foundation and NIH DK37332 to SRP We are grateful to Drs Christopher Wassif and Forbes Denny Porter (NIH) for independent confirmation of the mass spectrometry results and Dr Paul Saftig for providing MEF cells lacking LAMP1/LAMP2. Work at the Vincent Coates Foundation Mass Spectrometry Laboratory, Stanford University, was supported in part by NIH P30 CA124435.

## Additional information

#### Competing interests

SRP: Reviewing editor, *eLife*. The other author declares that no competing interests exist.

#### Funding

| Funder | Grant reference number | Author |
| --- | --- | --- |
| Ara Parseghian Medical Research Foundation | | Jian Li<br>Suzanne R Pfeffer |
| National Institute of Diabetes and Digestive and Kidney Diseases | DK37332 | Suzanne R Pfeffer |
| National Institutes of Health | P30 CA124435 | Suzanne R Pfeffer |

The funders had no role in study design, data collection and interpretation, or the decision to submit the work for publication.

#### Author contributions

JL, Conception and design, Acquisition of data, Analysis and interpretation of data, Drafting or revising the article; SRP, Conception and design, Analysis and interpretation of data, Drafting or revising the article

**Author ORCIDs**

Suzanne R Pfeffer, http://orcid.org/0000-0002-6462-984X

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
