## [Decision Letter]

Thank you for submitting your article "Lysosomal membrane glycoproteins bind cholesterol and contribute to lysosomal cholesterol export" for consideration by *eLife*. Your article has been favorably evaluated by Vivek Malhotra (Senior Editor) and three reviewers, one of whom, Chris G Burd (Reviewer #1), is a member of our Board of Reviewing Editors, and another one is Frederick Maxfield (Reviewer #2).

The reviewers have discussed the reviews with one another and the Reviewing Editor has drafted this decision to help you prepare a revised submission.

Summary:

Li and Pfeffer have investigated cholesterol binding by lysosomal LAMP proteins. They report that soluble, secreted fragments of the lumenal portions of LAMP1 and LAMP2 exhibit specific binding of cholesterol. Specificity of binding is suggested by saturation binding kinetics, competition experiments, and point mutagenesis. Full-length, endogenous LAMP2 is shown to co-immunoprecipitate with GFP-tagged NPC1 and NPC2 expressed by transfection and this association is diminished by treatment of cells to reduce late endosome cholesterol. Co-IP of LAMP2 and NPC1 is correlated with oligomeric state of LAMP1, where cholesterol promotes dissociation of LAMP1 oligomers and results in a decrease in the amount of LAMP2-NPC1 complex. Finally, the authors show that mutations in the lumenal domain of LAMP2 that ablate cholesterol binding do not support NPC1-mediated efflux of cholesterol from the lysosome.

Essential revisions:

The reviewers agreed that your findings that LAMP proteins bind to cholesterol and promote NPC1,2-mediated cholesterol export from the lysosome are potentially impactful for the field. However, four major concerns, listed below, were raised with several aspects of the work that must be definitively addressed in order to further advance the paper at *eLife*. While the reviewers indicated their willingness to recommend consideration of a revised manuscript by *eLife*, given the fundamental nature of several concerns, they also recommended that a revised manuscript should be evaluated by each reviewer.

1) The affinities and specificities of cholesterol binding to LAMP proteins is not adequately demonstrated.

It is concluded that 3H-cholesterol binds to LAMP2 with an affinity of Km ~ 5 µM. Given that the solubility of cholesterol in water is less than 100 nM, this result raises several critical questions. First, it is possible that a more soluble contaminant (e.g., an oxysterol) is responsible for the competition. Another possibility is that the cholesterol is in some type of micelle with the NP40 detergent. In either case, it is hard to know how to evaluate the concentration of cholesterol in these experiments. The authors need to elucidate the form of cholesterol that is bound by LAMP proteins.

The reviewers noted that the one-point competition experiments in Figure 1 are suggestive of specificity, but they agreed that it is essential to examine other structurally similar sterols (e.g., lanosterol), instead of cholesterol sulfate, which is more soluble than cholesterol. In addition, the competition data needs to show concentration curves of unlabeled competitor sterols.

2) The lipid components of the membranes from which LAMP2 is purified and bound lipids identified (Figure 2) must be determined. It is key to evaluate the co-purification data in light of the abundances of cholesterol versus other abundant lipids in lysosomal membranes such as phospholipids and sphingolipids. 7-β-hydroxy cholesterol and 26-hydroxy-cholesterol are present at trace levels in cells when compared to cholesterol, and are not appropriate controls for this experiment.

3) The conclusions that NPC1 and LAMPs bind each other and that this is regulated by cholesterol-dependent changes in oligomeric state of LAMP require additional support. Regarding the LAMP2-NPC1 binding experiment shown in Figure 3, data should be shown for reactions carried out in the absence of cholesterol. It was noted that the primary data showing cholesterol dependence of binding shown in Figure 3 was not shown and further, that the approach does not afford sufficient control of cholesterol level to firmly support the authors' conclusion that LAMP2 and NPC1 do not bind in the absence of cholesterol.

4) The reviewers had difficulty envisioning the role of LAMP proteins in NPC1,2-mediated cholesterol export from the lysosome as you suggest in the manuscript, and this requires serious consideration in the presentation of the proposed model. Specifically, previous work has suggested that NPC2 and NPC1-NTD bind cholesterol with opposite orientation of the hydroxyl, which would allow cholesterol to slide from one pocket to the other. If LAMPs and NPC1-NTD have the same orientation, this will not work; cholesterol would remain bound to LAMP. Related to this point, one reviewer raised the point that LAMPs seem better suited to be cholesterol sensors rather than "reservoirs". Compared to the capacity of the lysosome membrane to harbor cholesterol, are there really sufficient numbers of LAMP proteins per lysosome to constitute a "reservoir?"

[Editors' note: a revised version of this study was rejected after a second round of peer review, but the authors resubmitted for consideration, and the new manuscript was deemed suitable for publication.]

Thank you for submitting your work entitled "Lysosomal membrane glycoproteins bind cholesterol and contribute to lysosomal cholesterol export" for consideration by *eLife*. Your article has been favorably evaluated by Vivek Malhotra (Senior Editor) and three reviewers, one of whom, Chris G Burd (Reviewer #1), is a member of our Board of Reviewing Editors, and another one is Frederick Maxfield (Reviewer #2).

Our decision has been reached after consultation between the reviewers. Based on these discussions and the individual reviews below, we regret to inform you that your work will not be considered further for publication in *eLife*.

Overall, the reviewers found this to be an interesting and potentially important study. However, lingering concerns with several fundamental aspects of the study drove the editor's decision. Regarding cholesterol binding by LAMP proteins, the reviewers acknowledged the encouraging data that supports this conclusion, but there remains concern with the methodology, including a suggestion for more rigorous controls for non-specific binding. There was agreement amongst the reviewers that a well-supported and fundamental advance regarding the mechanism of cholesterol export has yet to emerge the experiments that probed interactions between LAMP2 and NPC1 (and possibly NPC2), and also the data presented regarding LAMP2 oligomerization.

*Reviewer #1:*

The authors have responded to each of the four major issues that were raised during the initial review. Each is addressed below.

"1) The affinities and specificities of cholesterol binding to LAMP proteins is not adequately demonstrated."

New competition binding data (Figure 2) over a 50-fold concentration range (I think – see below) show that 25-hydroxycholesterol and 7-ketocholesterol, but not cholesterol sulfate or lanosterol, compete with cholesterol for binding to secreted LAMP2 lumenal domain. The data, particularly for lanosterol, support the authors' conclusion that the 3β position of cholesterol is recognized by LAMP2.

The methods section needs to be updated. It is stated that competition experiments were done only in the presence of 30 μM unlabeled sterol.

"2) The lipid components of the membranes from which LAMP2 is purified and bound lipids identified (Figure 2) must be determined."

The authors point out that the material that was analyzed was secreted LAMP2, not material extracted from a membrane. I'm sorry – my mistake for not appreciating this previously. This, and the additional mass spectra analyses provide further support that cholesterol is specifically recognized by LAMP2.

"3) The conclusions that NPC1 and LAMPs bind each other and that this is regulated by cholesterol-dependent changes in oligomeric state of LAMP require additional support."

These concerns have been addressed with additional experiments.

The evidence presented in support of the conclusion that LAMP (LAMP1, in this case) dimerizes as a result of cholesterol depletion is an increase in the abundance of a high molecular weight species, consistent with a dimer, on a gel by anti-LAMP1 blot of cell extracts. Dimerization is one possibility. It could also be explained by an association with another protein, a conformational change, etc. While the magnitude in the increase in the putative LAMP1 dimer is statistically significant, it is a small proportion of the total LAMP1.

The IP in Figure 4 controls for co-IP of LAMP2 and NPC1-GFP, two integral membrane proteins, using just soluble GFP. A more appropriate and convincing control for specificity would be an unrelated GFP-tagged integral membrane protein.

The authors now report that binding of purified LAMP2 and NPC1 N-term domain proteins is not affected by the presence or absence of cholesterol, though co-IP of the full length proteins from cells is diminished when cells are incubated with cyclodextrin. The authors also report that LAMP proteins are present in approx. 10-fold excess of NPC1, and that binding between the luminal domains of LAMP2 and NPC1 in vitro are not affected by cholesterol occupancy. Given the 5 nM KD affinity of the interaction measured in vitro, it seems most likely to me that they would be associated constitutively (meaning, in the presence of absence of cholesterol) in the lysosome membrane.

Overall, with regards to the mechanism, it's unclear to me what the data is telling us. Regardless of the mechanism, the data do support, in my opinion, the conclusion that LAMP2 plays a role in cholesterol efflux from the lysosome and that this is correlated with cholesterol binding to LAMP2.

"4) The reviewers had difficulty envisioning the role of LAMP proteins in NPC1,2-mediated cholesterol export from the lysosome as you suggest in the manuscript, and this requires serious consideration in the presentation of the proposed model."

With the clarification of the manner in which the authors use "reservoir," along with the determination of LAMP protein abundance, I agree that it is plausible that LAMP proteins provide a physiologically relevant "reservoir" of cholesterol to be a component of the NPC1- and NPC2-mediated lysosomal cholesterol efflux pathway.

*Reviewer #2:*

In general the authors have addressed my major concerns. However, I still have some questions and concerns that need to be addressed.

1) I don't understand the comment in the fourth paragraph of the Results and Discussion section. I don't see how having 25-OH cholesterol will improve the solubility of cholesterol and increase its binding.

2) The treatment of wild type cells with cyclodextrin is confusing to me. Does it actually reduce cholesterol in the late endosomes/lysosomes? Without showing this, the effect on LAMP1 is hard to interpret. I am also concerned about the health of cells in LPDS and 1 mM CD for 24 hours.

3) In the eighth paragraph of the Results and Discussion section. This explanation does not make sense to me. Only a small fraction of the LAMP is dimerized. There is still abundant monomer, which could interact with NPC1. I would recommend deleting the CD treatment and the LAMP dimerization. It is not key to the paper.

*Reviewer #3:*

In this revision, the authors have attempted to address the concerns raised with the cholesterol binding data and with their claims of cholesterol-regulated behavior of LAMP proteins and interaction with NPC proteins. Unfortunately, almost all of my concerns remain.

Specific points:

My previous review: “The assay showing direct binding of 3H-cholesterol to LAMP2 has no specificity controls (Figure 1). Since 3H-cholesterol binding to LAMP2 shows weak affinity (Km ~ 5 µM), it is imperative to show that some other hydrophobic 3H-ligand does not bind to LAMP2 at these high concentrations where hydrophobic ligands often precipitate. The one-point competition experiments in Figure 1 are suggestive of specificity, but the authors should try lanosterol (or some other structurally similar sterol) as a negative control, instead of cholesterol sulfate which is more soluble than cholesterol. The competition data also needs to show concentration curves of unlabeled competitor sterols. The authors' claim of similar binding of cholesterol to LAMP2 and NPC1-NTD is misleading since they compare saturation stoichiometries, not the more relevant affinity parameters. To rule out non-specific binding of insoluble cholesterol to LAMP2, careful controls are needed”.

The authors now say that they carried out binding reactions in the presence of sub-cmc concentrations of detergent. This sounds good, but the Methods section does not say anything about this. The only place the detergent is mentioned is in a new paragraph in the Results, but no details are given. All the binding data in Figure 1 is identical to what was in the first submission. So, did the authors redo these with sub-cmc detergent and get the same results? In any case, even if the authors had merely not reported their experimental procedures accurately in the original submission, none of the concerns previously raised have been addressed.

A) The key data is in Figure 1 which shows a saturation binding curve of 3H-cholesterol to LAMP2 that shows very weak affinity – Km = 5-10 µM. This solitary curve is hard to interpret without carrying out the same dose curve in the presence of excess unlabeled cholesterol, which would indicate the background non-specific binding. Also, they need to show that some other 3H-sterol ligand does not bind LAMP2 in the same experiment. They claim later that 25-HC competes and lanosterol does not compete. Both of these sterols are available commercially in tritiated form. 3H-lanosterol would have been a great control for specificity.

B) The next key pieces of data are in 1G and 1H. Again, these panels are identical to the original submission, and once again it is not clear how much 3H-cholesterol was used. From extrapolation using 1B, it looks like around 5 µM, but details like these are critical and need to be indicated. In any case, 1H shows competition by unlabeled cholesterol, but there is no negative control with a non-competitor like cholesterol sulfate. The authors now add competition curves by other unlabeled sterols in Figure 2. Unfortunately, these cannot be judged because once again it is not clear how much 3H-cholesterol was used and the raw binding values that were normalized to 1.0 are not indicated. The authors should use cpm units so that we can compare with Figure 1 data. Also, every competition curve in Figure 2 requires a positive control and a negative control in the same experiment to be able to judge anything. They show competition curve for unlabeled cholesterol out to 15 µM in Figure 1, but all the other competition curves are upto 50 µM – what happens with unlabeled cholesterol at these concentrations. Based on the cholesterol data, the competitors should be judged by their ability to compete at 15 µM. Only 7-ketocholesterol seems to be a strong competitor. Also, why did they not do competition curves for 24-hydroxycholesterol, which according to Figure 1 was the strongest competitor? How about epicholesterol, where the 3-hydroxyl region is modified in a manner more subtle than cholesterol sulfate. The claim that the 3β-hydroxyl group of cholesterol enters the binding pocket cannot be made from these data.

C) The mass spec data showing cholesterol is present in purifications could be due to the high concentrations of cholesterol, but not their control sterols, in the serum-rich media into which the proteins were secreted. Again, no details of the growth conditions are given so that we can judge these results.

My previous review: “The claim that LAMP proteins oligomerize when cholesterol is limiting and associate with NPC1 when cholesterol is available is not supported by the data of Figure 3. The association of NPC1 with LAMP2 is shown by IP in Figure 3, but no cholesterol dependence for this interaction is shown (the raw data for the bar graph in 3B is not shown). Instead, the interaction of NPC1 with LAMP2 in the presence of cholesterol is shown by a different method in Figure 3, but this experiment does not show what happens in the absence of cholesterol! Figure 3 shows a partial effect of cholesterol on LAMP1 dimerization, but not on LAMP2 oligomers. The key lane on the gel in Figure 3 is shown in isolation without any controls, so the significance of higher density of dimer is not clear. The quantification needs to be done on a gel from a single experiment with LAMP2.”

The authors had previously claimed cholesterol dependency of NPC1-LAMP2 interaction, but had not shown the data in a single experiment. They now say that there is no cholesterol dependence, but again do not show the data!

The IP data showing interaction between LAMP2 and NPC1-GFP (a membrane protein) uses GFP (soluble protein) as a control. They would need to use another lysosomal membrane protein as a control to rule in specificity. Again, they claim that LAMP1 dimerizes using a gel where the key lane is shown in isolation without any controls (Figure 4), so the significance of the slightly higher density of the dimer is not clear. The quantification needs to be done on a gel from a single experiment with LAMP1.

Eskelinen et al., 2004 and Schneede et al., 2011 have already highlighted the roles of Lamp2 in cholesterol transport, the key advance in this study is to attempt to show a direct interaction between cholesterol and LAMP proteins and to get at the mechanism. Neither of these points are supported by their data, and as such I do not think this paper makes a significant contribution to this problem.

---

## [Author Response]

*Essential revisions:*

*The reviewers agreed that your findings that LAMP proteins bind to cholesterol and promote NPC1,2-mediated cholesterol export from the lysosome are potentially impactful for the field. However, four major concerns, listed below, were raised with several aspects of the work that must be definitively addressed in order to further advance the paper at eLife. While the reviewers indicated their willingness to recommend consideration of a revised manuscript by eLife, given the fundamental nature of several concerns, they also recommended that a revised manuscript should be evaluated by each reviewer.*

*1) The affinities and specificities of cholesterol binding to LAMP proteins is not adequately demonstrated.*

*It is concluded that 3H-cholesterol binds to LAMP2 with an affinity of Km ~ 5 µM. Given that the solubility of cholesterol in water is less than 100 nM, this result raises several critical questions. First, it is possible that a more soluble contaminant (e.g., an oxysterol) is responsible for the competition. Another possibility is that the cholesterol is in some type of micelle with the NP40 detergent. In either case, it is hard to know how to evaluate the concentration of cholesterol in these experiments. The authors need to elucidate the form of cholesterol that is bound by LAMP proteins.*

We agree with the reviewers that cholesterol affinities are difficult to measure and we clarified the text to explain this. We have used precisely the conditions employed by Brown & Goldstein in their studies of cholesterol binding to NPC1 protein.

New text: “Binding reactions were carried out in the presence of sub-critical micelle concentration concentrations of Nonidet P40 detergent to help solubilize the cholesterol (Infante et al., 2008). Under these conditions, most of the cholesterol remains in a mixed micelle of cholesterol and detergent and is still poorly soluble. Thus, the apparent affinity for cholesterol is likely to be tighter than the curves indicate, as the amounts added do not reflect the concentration of free cholesterol that is available for binding.”

*The reviewers noted that the one-point competition experiments in Figure 1 are suggestive of specificity, but they agreed that it is essential to examine other structurally similar sterols (e.g., lanosterol), instead of cholesterol sulfate, which is more soluble than cholesterol. In addition, the competition data needs to show concentration curves of unlabeled competitor sterols.*

To better characterize the binding pocket we have now added a new Figure 2with complete binding profiles for cholesterol sulfate, 25-hydroxycholesterol, lanosterol and 7-ketocholesterol. Only 7-ketocholesterol and 25- hydroxycholesterol compete, consistent with our conclusion that the 3β-hydroxyl group of cholesterol enters the binding pocket.

*2) The lipid components of the membranes from which LAMP2 is purified and bound lipids identified (Figure 2) must be determined. It is key to evaluate the co-purification data in light of the abundances of cholesterol versus other abundant lipids in lysosomal membranes such as phospholipids and sphingolipids. 7-β-hydroxy cholesterol and 26-hydroxy-cholesterol are present at trace levels in cells when compared to cholesterol, and are not appropriate controls for this experiment.*

The protein is purified from secretions and not from membranes. A request for full lipid analysis is far beyond what Brown & Goldstein needed for NPC1 as a cholesterol binder. We added a new chromatogram to show that lanosterol, 24-, and 25-hydroxycholesterol also fail to co-chromatograph with the eluted material. Importantly, the mass peaks perfectly correspond to cholesterol or a derivative thereof, rather than a phospholipid or sphingolipid. Also, the binding specificity seen by competition (new Figure 2) is consistent with this. Finally, we sent our samples to Denny Porter and Chris Wassif at NIH who independently got the same profile that we did; they found no evidence of any phospholipids or special sterols.

*3) The conclusions that NPC1 and LAMPs bind each other and that this is regulated by cholesterol-dependent changes in oligomeric state of LAMP require additional support. Regarding the LAMP2-NPC1 binding experiment shown in Figure 3, data should be shown for reactions carried out in the absence of cholesterol. It was noted that the primary data showing cholesterol dependence of binding shown in Figure 3 was not shown and further, that the approach does not afford sufficient control of cholesterol level to firmly support the authors' conclusion that LAMP2 and NPC1 do not bind in the absence of cholesterol.*

We repeated the in vitro MST experiments with and without cholesterol and find that the N-terminal domain of NPC1 binds LAMP2 under either condition and now report this. We also added additional data in support of decreased interaction of LAMP2 with NPC1-GFP in cells with the addition of the actual gels (new Figure 4 panel B). Thus, although the NPC1 N-terminal domain binds LAMP2 whether or not that NPC1 domain is occupied, the total proteins don’t interact as well when cholesterol is limiting in lysosomes in cells.

*4) The reviewers had difficulty envisioning the role of LAMP proteins in NPC1,2-mediated cholesterol export from the lysosome as you suggest in the manuscript, and this requires serious consideration in the presentation of the proposed model. Specifically, previous work has suggested that NPC2 and NPC1-NTD bind cholesterol with opposite orientation of the hydroxyl, which would allow cholesterol to slide from one pocket to the other. If LAMPs and NPC1-NTD have the same orientation, this will not work; cholesterol would remain bound to LAMP. Related to this point, one reviewer raised the point that LAMPs seem better suited to be cholesterol sensors rather than "reservoirs". Compared to the capacity of the lysosome membrane to harbor cholesterol, are there really sufficient numbers of LAMP proteins per lysosome to constitute a "reservoir?"*

We actually went ahead and quantified the amount of LAMP and NPC1 proteins in cells using purified protein standards (new Figure 5—figure supplement 1). The 3D LAMP2 concentration in 0.3mM which is not insignificant; it will be higher in 2D environment. We also clarified the Discussion to explain more clearly how LAMP proteins may hold soluble cholesterol for NPC2-mediated transfer to NPC1.

“…How might LAMP2’s cholesterol binding site contribute to cholesterol export? Current models suggest that the soluble NPC2 protein binds cholesterol from the internal membranes of lysosomes and delivers it to NPC1 at the limiting membrane of this compartment (Kwon et al., 2009). […] In future work, it will be important to elucidate precisely how LAMP2 interacts with both NPC2 and NPC1 to facilitate cholesterol export from lysosomes and how cholesterol binding contributes to LAMP2’s other cellular roles.”

[Editors' note: what now follows is the decision letter after the authors resubmitted for further consideration.]

*Overall, the reviewers found this to be an interesting and potentially important study. However, lingering concerns with several fundamental aspects of the study drove the editor's decision. Regarding cholesterol binding by LAMP proteins, the reviewers acknowledged the encouraging data that supports this conclusion, but there remains concern with the methodology, including a suggestion for more rigorous controls for non-specific binding. There was agreement amongst the reviewers that a well-supported and fundamental advance regarding the mechanism of cholesterol export has yet to emerge the experiments that probed interactions between LAMP2 and NPC1 (and possibly NPC2), and also the data presented regarding LAMP2 oligomerization.*

*Reviewer #1:*

*The authors have responded to each of the four major issues that were raised during the initial review. Each is addressed below.*

*"1) The affinities and specificities of cholesterol binding to LAMP proteins is not adequately demonstrated."*

*New competition binding data (Figure 2) over a 50-fold concentration range (I think – see below) show that 25-hydroxycholesterol and 7-ketocholesterol, but not cholesterol sulfate or lanosterol, compete with cholesterol for binding to secreted LAMP2 lumenal domain. The data, particularly for lanosterol, support the authors' conclusion that the 3β position of cholesterol is recognized by LAMP2.*

Thank you. We have now added epicholesterol which is the best control for cholesterol specificity.

*The methods section needs to be updated. It is stated that competition experiments were done only in the presence of 30 μM unlabeled sterol.*

Corrected.

*"2) The lipid components of the membranes from which LAMP2 is purified and bound lipids identified (Figure 2) must be determined."*

*The authors point out that the material that was analyzed was secreted LAMP2, not material extracted from a membrane. I'm sorry – my mistake for not appreciating this previously. This, and the additional mass spectra analyses provide further support that cholesterol is specifically recognized by LAMP2.*

Thank you and we have importantly clarified that there was no cholesterol in the medium.

*"3) The conclusions that NPC1 and LAMPs bind each other and that this is regulated by cholesterol-dependent changes in oligomeric state of LAMP require additional support."*

*These concerns have been addressed with additional experiments.*

*The evidence presented in support of the conclusion that LAMP (LAMP1, in this case) dimerizes as a result of cholesterol depletion is an increase in the abundance of a high molecular weight species, consistent with a dimer, on a gel by anti-LAMP1 blot of cell extracts. Dimerization is one possibility. It could also be explained by an association with another protein, a conformational change, etc. While the magnitude in the increase in the putative LAMP1 dimer is statistically significant, it is a small proportion of the total LAMP1.*

We have removed this data.

*The IP in Figure 4 controls for co-IP of LAMP2 and NPC1-GFP, two integral membrane proteins, using just soluble GFP. A more appropriate and convincing control for specificity would be an unrelated GFP-tagged integral membrane protein.*

We have added the control lysosomal integral membrane protein, MCOLN1 and showed that it does not bind LAMP2.

*The authors now report that binding of purified LAMP2 and NPC1 N-term domain proteins is not affected by the presence or absence of cholesterol, though co-IP of the full length proteins from cells is diminished when cells are incubated with cyclodextrin. The authors also report that LAMP proteins are present in approx. 10-fold excess of NPC1, and that binding between the luminal domains of LAMP2 and NPC1 in vitro are not affected by cholesterol occupancy. Given the 5 nM KD affinity of the interaction measured in vitro, it seems most likely to me that they would be associated constitutively (meaning, in the presence of absence of cholesterol) in the lysosome membrane.*

Agreed (for the *N-terminus* which we measured in vitro, ± cholesterol); but in cells, the N-terminus may bind other things such as the rest of NPC1. The full length protein does show a difference in Co-IPs under different cholesterol concentrations.

*Overall, with regards to the mechanism, it's unclear to me what the data is telling us. Regardless of the mechanism, the data do support, in my opinion, the conclusion that LAMP2 plays a role in cholesterol efflux from the lysosome and that this is correlated with cholesterol binding to LAMP2.*

Thank you; the field needs new models to explain our findings.

*"4) The reviewers had difficulty envisioning the role of LAMP proteins in NPC1,2-mediated cholesterol export from the lysosome as you suggest in the manuscript, and this requires serious consideration in the presentation of the proposed model."*

*With the clarification of the manner in which the authors use "reservoir," along with the determination of LAMP protein abundance, I agree that it is plausible that LAMP proteins provide a physiologically relevant "reservoir" of cholesterol to be a component of the NPC1- and NPC2-mediated lysosomal cholesterol efflux pathway.*

Thank you.

*Reviewer #2:*

*In general the authors have addressed my major concerns. However, I still have some questions and concerns that need to be addressed.*

*1) I don't understand the comment in the fourth paragraph of the Results and Discussion section. I don't see how having 25-OH cholesterol will improve the solubility of cholesterol and increase its binding.*

All the binding experiments use sub-CMC Triton to solubilize the sterol as worked out by Brown and Goldstein for the N-terminal domain of NPC1 (Infante et al. 2008 JBC 283, 1052). 25H is slightly more soluble than regular cholesterol, and all sterols come from mixed micelles onto LAMP protein in the binding reactions. Thus, there will be mixed micelles of 25H and regular cholesterol. We tried to clarify this in the text; (Infante et al. 2008 saw the same phenomena in the above paper in 3 figures).

*2) The treatment of wild type cells with cyclodextrin is confusing to me. Does it actually reduce cholesterol in the late endosomes/lysosomes?*

Yes, it does and it can cure the NPC phenotype of cholesterol accumulation (Maxfield (PNAS 107, 5477) and Brown/Goldstein now cited). This is now more clearly explained in the text.

*Without showing this, the effect on LAMP1 is hard to interpret. I am also concerned about the health of cells in LPDS and 1 mM CD for 24 hours.*

Brown and Goldstein showed that release of cholesterol from lysosomes, assayed by formation of cholesteryl esters, was maximal at 6 hours of treatment with 0.1% cyclodextrin (~1mM); they also assayed 24hours in their time course (Abi-Mosleh et al. 2009 PNAS). We saw normal growth rate and no cell death after 24 hours. The text was modified to clarify this point, and added a toxicology reference that supports our findings.

*3) In the eighth paragraph of the Results and Discussion section. This explanation does not make sense to me. Only a small fraction of the LAMP is dimerized. There is still abundant monomer, which could interact with NPC1. I would recommend deleting the CD treatment and the LAMP dimerization. It is not key to the paper.*

Deleted as requested; such assays always capture only a small percent of oligomers in any case.

*Reviewer #3:*

*In this revision, the authors have attempted to address the concerns raised with the cholesterol binding data and with their claims of cholesterol-regulated behavior of LAMP proteins and interaction with NPC proteins. Unfortunately, almost all of my concerns remain.*

*Specific points:*

*My previous review: “The assay showing direct binding of 3H-cholesterol to LAMP2 has no specificity controls (Figure 1). Since 3H-cholesterol binding to LAMP2 shows weak affinity (Km ~ 5 µM), it is imperative to show that some other hydrophobic 3H-ligand does not bind to LAMP2 at these high concentrations where hydrophobic ligands often precipitate”.*

Done (see below).

*“The one-point competition experiments in Figure 1 are suggestive of specificity, but the authors should try lanosterol (or some other structurally similar sterol) as a negative control, instead of cholesterol sulfate which is more soluble than cholesterol. The competition data also needs to show concentration curves of unlabeled competitor sterols. The authors' claim of similar binding of cholesterol to LAMP2 and NPC1-NTD is misleading since they compare saturation stoichiometries, not the more relevant affinity parameters. To rule out non-specific binding of insoluble cholesterol to LAMP2, careful controls are needed”.*

*The authors now say that they carried out binding reactions in the presence of sub-cmc concentrations of detergent. This sounds good, but the Methods section does not say anything about this. The only place the detergent is mentioned is in a new paragraph in the Results, but no details are given. All the binding data in Figure 1 is identical to what was in the first submission. So, did the authors redo these with sub-cmc detergent and get the same results? In any case, even if the authors had merely not reported their experimental procedures accurately in the original submission, none of the concerns previously raised have been addressed.*

We have always used conditions established by Brown and Goldstein for NPC1 and explain that more clearly here. We thank the referee for helping us make this clearer.

*A) The key data is in Figure 1 which shows a saturation binding curve of 3H-cholesterol to LAMP2 that shows very weak affinity – Km = 5-10 µM. This solitary curve is hard to interpret without carrying out the same dose curve in the presence of excess unlabeled cholesterol, which would indicate the background non-specific binding. Also, they need to show that some other 3H-sterol ligand does not bind LAMP2 in the same experiment. They claim later that 25-HC competes and lanosterol does not compete. Both of these sterols are available commercially in tritiated form. 3H-lanosterol would have been a great control for specificity.*

*B) The next key pieces of data are in 1G and 1H. Again, these panels are identical to the original submission, and once again it is not clear how much 3H-cholesterol was used. From extrapolation using 1B, it looks like around 5 µM, but details like these are critical and need to be indicated. In any case, 1H shows competition by unlabeled cholesterol, but there is no negative control with a non-competitor like cholesterol sulfate. The authors now add competition curves by other unlabeled sterols in Figure 2. Unfortunately, these cannot be judged because once again it is not clear how much 3H-cholesterol was used and the raw binding values that were normalized to 1.0 are not indicated. The authors should use cpm units so that we can compare with Figure 1 data. Also, every competition curve in Figure 2 requires a positive control and a negative control in the same experiment to be able to judge anything. They show competition curve for unlabeled cholesterol out to 15 µM in Figure 1, but all the other competition curves are upto 50 µM – what happens with unlabeled cholesterol at these concentrations. Based on the cholesterol data, the competitors should be judged by their ability to compete at 15 µM. Only 7-ketocholesterol seems to be a strong competitor. Also, why did they not do competition curves for 24-hydroxycholesterol, which according to Figure 1 was the strongest competitor? How about epicholesterol, where the 3-hydroxyl region is modified in a manner more subtle than cholesterol sulfate. The claim that the 3β-hydroxyl group of cholesterol enters the binding pocket cannot be made from these data.*

A _3_H-lanosterol control would have cost $2500. Instead we used the excellent suggestion of epicholesterol which wonderfully does not bind! (new Figure 2, controls included). Thank you so much for this excellent suggestion. As noted by the referee, at high sterol concentrations, it is not possible to add enough competitor to fully block binding as there are significant solubility issues. Nevertheless, we have now made much more clear how the binding was done, how much cholesterol is present, and added new data for 25-hydroxycholesterol ± cold competitor (new Figure 1). In summary, sterol binding is indeed specific and likely tighter than the apparent Kd due to solubility issues.

*C) The mass spec data showing cholesterol is present in purifications could be due to the high concentrations of cholesterol, but not their control sterols, in the serum-rich media into which the proteins were secreted. Again, no details of the growth conditions are given so that we can judge these results.*

Our error – the protein was purified using Freestyle 293 medium lacking any cholesterol, thank you for catching our omission.

*My previous review: “The claim that LAMP proteins oligomerize when cholesterol is limiting and associate with NPC1 when cholesterol is available is not supported by the data of Figure 3. The association of NPC1 with LAMP2 is shown by IP in Figure 3, but no cholesterol dependence for this interaction is shown (the raw data for the bar graph in 3B is not shown). Instead, the interaction of NPC1 with LAMP2 in the presence of cholesterol is shown by a different method in Figure 3, but this experiment does not show what happens in the absence of cholesterol! Figure 3 shows a partial effect of cholesterol on LAMP1 dimerization, but not on LAMP2 oligomers. The key lane on the gel in Figure 3 is shown in isolation without any controls, so the significance of higher density of dimer is not clear. The quantification needs to be done on a gel from a single experiment with LAMP2”.*

*The authors had previously claimed cholesterol dependency of NPC1-LAMP2 interaction, but had not shown the data in a single experiment. They now say that there is no cholesterol dependence, but again do not show the data!*

*The IP data showing interaction between LAMP2 and NPC1-GFP (a membrane protein) uses GFP (soluble protein) as a control. They would need to use another lysosomal membrane protein as a control to rule in specificity. Again, they claim that LAMP1 dimerizes using a gel where the key lane is shown in isolation without any controls (Figure 4), so the significance of the slightly higher density of the dimer is not clear. The quantification needs to be done on a gel from a single experiment with LAMP1.*

We have added the control lysosomal integral membrane protein, MCOLN1 and

showed that it does not bind LAMP2. We did show the data for the quantitation and removed the crosslinking to avoid any issues.

*Eskelinen et al., 2004 and Schneede et al., 2011 have already highlighted the roles of Lamp2 in cholesterol transport, the key advance in this study is to attempt to show a direct interaction between cholesterol and LAMP proteins and to get at the mechanism. Neither of these points are supported by their data, and as such I do not think this paper makes a significant contribution to this problem.*

We feel strongly that the demonstration of direct cholesterol binding to LAMP2, its tight interaction with NPC1 in vitro and in cells, and the correlation between cholesterol binding capacity and phenotype rescue of LAMP depleted cells, represent a very unexpected and significant advance in our understanding of what role this protein family may play. Previous work provided no hints as to why LAMP1/2 depletion interfered with cholesterol export from lysosomes. Our work provides entirely new models that can be tested and studied further.